

# SPANet: Generalized permutationless set assignment for particle physics using symmetry preserving attention

Alexander Shmakov[1*], Michael J. Fenton[1†], Ta-Wei Ho[2],
Shih-Chieh Hsu[3], Daniel Whiteson[1] and Pierre Baldi[1]

**1** Department of Computer Science, University of California Irvine, Irvine, CA
**2** Department of Physics and Astronomy, National Tsing Hua University, Hsinchu City, Taiwan
**3** Department of Physics and Astronomy, University of Washington, Seattle, WA

⋆ ashmakov@uci.edu , † mjfenton@uci.edu
These two authors have contributed equally

## Abstract

The creation of unstable heavy particles at the Large Hadron Collider is the most direct way to address some of the deepest open questions in physics. Collisions typically produce variable-size *sets* of observed particles which have inherent ambiguities complicating the assignment of observed particles to the decay products of the heavy particles. Current strategies for tackling these challenges in the physics community ignore the physical symmetries of the decay products and consider all possible assignment permutations and do not scale to complex configurations. Attention based deep learning methods for sequence modelling have achieved state-of-the-art performance in natural language processing, but they lack built-in mechanisms to deal with the unique symmetries found in physical set-assignment problems. We introduce a novel method for constructing symmetry-preserving attention networks which reflect the problem's natural invariances to efficiently find assignments without evaluating all permutations. This general approach is applicable to arbitrarily complex configurations and significantly outperforms current methods, improving reconstruction efficiency between 19% - 35% on typical benchmark problems while decreasing inference time by two to five orders of magnitude on the most complex events, making many important and previously intractable cases tractable. A full code repository containing a general library, the specific configuration used, and a complete dataset release, are available at https://github.com/Alexanders101/SPANet



# 1   Introduction

Many of the most important mysteries in modern physics, such as the nature of dark matter or a quantum description of gravity, can be studied through high-energy particle collisions. The frontier of such research is at the Large Hadron Collider (LHC) [1], which smashes protons together at energies that reproduce the conditions just after the Big Bang, thereby creating heavy, unstable, particles that could provide critical clues to unravel these mysteries. But these unstable particles are too short-lived to be studied directly, and can only be observed through the patterns of the particles into which they decay. A large fraction of these decay products lead to *jets*, streams of collimated particles which are virtually indistinguishable from each other. However, jets may also be produced through many other physical processes, and the ambiguity of which of these jets originates from which of the decay products obscures the decay pattern of the heavy particles, crippling the ability of physicists to extract vital scientific information from their data.

Reconstructing such events consists of assigning specific labels to a *variable-size set* of observed jets, each represented by a fixed-size vector of physical measurements of the jet. Each label represents a decay product of the heavy particle and must be uniquely assigned to one of the jets if the mass and momentum of the heavy particle is to be measured. Current solutions consider all possible assignment permutations, an ineffective strategy for which the combinatorial computation cost grows so rapidly with the number of jets that they are rendered unusable in particle collisions with more than a handful of jets.

Event reconstruction may be viewed as a specific case of a general problem we refer to as *set assignment*. Formally, given a label set $T = \{t_1, t_2, \ldots, t_C\}$ and input set $X = \{x_1, x_2, \ldots, x_N\}$, with $N \geq C$, the goal is to assign every label $t \in T$ to an element $x \in X$. Set assignment is fundamental to many problems in the physical sciences including cosmology [2,3] and high energy physics [4,5]. Several other problems may be viewed as variants of set assignment where labels must be *uniquely* assigned, including learning-to-rank [6], where labels correspond to numerical ranks, and permutation learning [7], where sets of objects must be correctly ordered. These unique assignment problems are critical to a variety of applications such as recommender systems [8], anomaly detection [9], and image-set classification [10]. Event reconstruction presents a particularly challenging variant of set assignment where: (1) the input consists of variable-size sets; (2) the labels must be assigned uniquely; and (3) additional label symmetries (described in Section 2), arising from the laws of physics, must be preserved. Many methods tackle each of these complexities individually, but no current methods effectively incorporate all of these constraints.

Several deep learning methods have been developed to handle variable-length sequences, fixed-size sets, and more recently even variable-size sets [11]. In particular, attention-based methods have achieved state-of-the-art results in natural language processing problems such as translation [12–15], where variable-length sequences are common. Among these methods, transformers [16] stand out as particularly promising for set assignment due to their fundamental invariance with respect to the order of the input sequence [11]. Transformers are especially effective at modeling variable-length sets because they can learn combinatorial relationships between set elements with a polynomial run-time.

Although transformers effectively encode input permutation invariance, they have no inherent mechanism for ensuring unique assignments or invariance with respect to general label symmetries. Techniques to imbue network architectures with general symmetries have been studied to design convolution networks operating on topological spaces [17–19]. However, these approaches focus primarily on invariances with respect to *input* transformations (e.g. rotations, translations), as opposed to invariances with respect to labels. We present a novel attention-based method which expands on the transformer to tackle the unique symmetries and challenges present in LHC event reconstruction.

## 2 Event Reconstruction at the LHC

The various detectors at the Large Hadron Collider measure particles produced in the high energy collisions of protons. In each collision event, heavy, unstable particles such as top quarks, Higgs-bosons, or $W-$ & $Z$-bosons may be created. These *resonance particles* decay too quickly ($< 10^{-20}$s) to be directly detected [20]. To study them, experimentalists must reconstruct them from their decay products, *partons*. From fundamental models of particle interactions, represented as Feynman diagrams (Figure 1), we know which partons to expect from each resonance. When these partons are *quarks* they appear in the detectors as *jets*, collimated streams of particles. However, collisions commonly produce many more jets than just those from the resonance particles. In order to reconstruct the resonance particles, the jets produced from the partons must then be identified. Event reconstruction reduces to uniquely assigning a collection of labels - the parent partons - to a collection of observed jets. We refer to this as the *jet-parton assignment* problem.

### 2.1 Symmetries

The essential difficulty of the task stems from the fact that the detector signature of jets from most types of partons are nearly indistinguishable. Jets are represented as a 4-dimensional

momentum vector called a *4-vector*, with one additional dimension which indicates whether the jet likely originated from a bottom quark, which can be identified somewhat reliably using multi-variate techniques (referred to as *b*-tagging), or a light[1] quark. The electric charge of the originating particle cannot be reliably deduced from a jet, such that quarks and anti-quarks give practically identical detector signatures. Jets are also not uniquely produced by quarks, but may also result from the production of gluons[2]. We refer to all of these peculiarities as *particle symmetries*.

Additionally, in some cases, the reconstruction task is insensitive to swapping labels. For example, while a *W* boson decays to a quark and anti-quark, inverting the labels leads to the same reconstructed *W* boson for most experiment setups. We refer to these lower-level invariances on the jet-labels as *jet symmetries*. Exploiting all of these symmetries is crucial for effective event reconstruction, especially in complex events with many jets where these invariances greatly reduce the number of possible jet assignments. Therefore, incorporating symmetries into reconstruction models may substantially simplify the modeling task. We refer to the complete specification of an event's particles and all of their associated symmetries as its *topology*.

## 2.2 Benchmark Events

We study event reconstruction for three common topologies, although the techniques generalize to arbitrary event topologies. The first is the production of a top/anti-top pair ($t\bar{t}$). Top quarks are very heavy, and decay so quickly that they are considered a resonance particle rather than a parton. Top quarks decay almost exclusively to a *b*-quark and a *W*-boson, which most commonly then decays to a further two light quarks (visualized in Figure 1a). $t\bar{t}$ production can therefore lead to six jets and is thus a canonical example of the jet-parton assignment problem[3] and is an extremely important task in LHC physics. Nonetheless, $t\bar{t}$ production leading to six jets is a comparatively under-explored signature, given the difficulty of the assignment task and copious production of $\geq 6$ jet events from processes which do not involve top quarks. We exploit the jet symmetry between the light quark jets from the *W*-bosons to aid us in finding solutions to this problem, as well as the particle symmetry between the top and anti-top.

We further study two more complex final states; top-quark-associated Higgs production

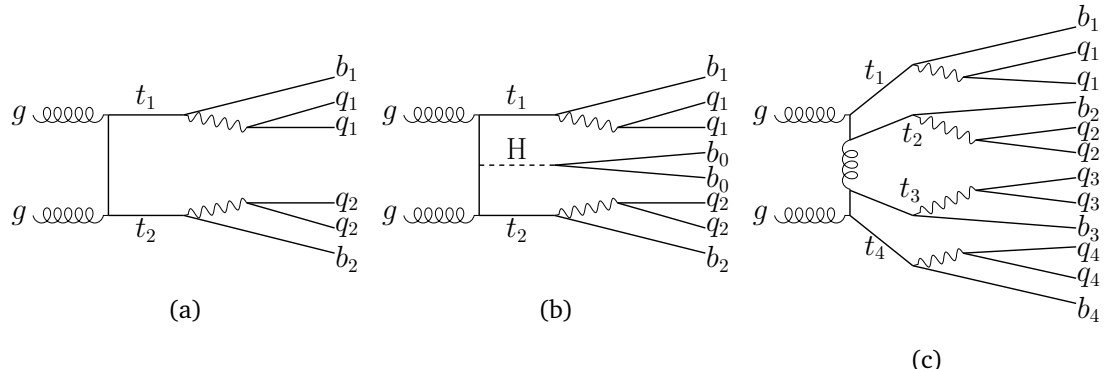

Figure 1: Feynman diagrams, a visual description of the decay of resonance particles into partons, for (a) $t\bar{t}$ events, (b) $t\bar{t}H$ events, and (c) $t\bar{t}t\bar{t}$ events.

---

[1]We define *light quark* to mean up, down, strange or charm in this context.

[2]The problem of quark/gluon discrimination is an active field of research [21–23], and adding such techniques as input to SPA-NET is a promising avenue for future work to improve performance.

[3]We restrict our discussion throughout to *all-jet* topologies, where all partons are quarks. We leave investigation of final states involving other partons, such as leptons and photons, to future work.

($t\bar{t}H$), and 4-top production ($t\bar{t}t\bar{t}$), as shown in Figures 1b and 1c respectively. The $t\bar{t}H$ process is of particular interest at the LHC as a direct probe of the top-Higgs Yukawa coupling, and is a rare process that has only recently been discovered by the ATLAS [24] and CMS [25,26] collaborations. However, $t\bar{t}H$ with the Higgs decaying to a pair of $b$-quarks is not the most sensitive channel due in part to the complexity of correctly reconstructing the events [27]. On top of the symmetries described for the $t\bar{t}$ case, we can further exploit the symmetry between the $b$-quarks from the Higgs for this problem.

The $t\bar{t}t\bar{t}$ process represents an even more difficult topology. There is strong evidence for the existence of this process from events including decays to leptons [28–31]. To the best of our knowledge, no attempt has been made to analyse the $t\bar{t}t\bar{t}$ process in the all-jet channel, an extremely complex topology involving the assignment of 12 partons. In this case, there is a 4-way symmetry between the top quarks, and 4 instances of symmetry between $W$-boson decays. This topology is then an extremely interesting stress test of set assignment methods, and represents a final state that has, to date, been impossible to fully reconstruct.

## 2.3 Baseline Methods

We implement the $\chi^2$ minimisation technique previously used by ATLAS [32,33] for jet-parton assignment in the $t\bar{t}$ process, which compares the masses of the reconstructed top and $W$ particles to their known values. Similar techniques have been used by CMS [34]. The $\chi^2$ for $t\bar{t}$ is defined as

$$\chi^2_{t\bar{t}} = \frac{(m_{b_1 q_1 q_1} - m_t)^2}{\sigma_t^2} + \frac{(m_{b_2 q_2 q_2} - m_t)^2}{\sigma_t^2} + \frac{(m_{q_1 q_1} - m_W)^2}{\sigma_W^2} + \frac{(m_{q_2 q_2} - m_W)^2}{\sigma_W^2}, \qquad (1)$$

where $m_{b_i q_i q_i}$ is the invariant mass of the jets in that permutation, and $\sigma_t$ and $\sigma_W$ are the widths of the resonances fitted in the dataset. In our datasets, described in Section 4, we find $\sigma_t = 28.8$ GeV and $\sigma_W = 18.7$ GeV using a Gaussian fit.

The $\chi^2$ method is an example of a permutation approach to set assignment, in which every possible jet permutation is explicitly tested to produce the highest scoring assignment. While effective, this method suffers from exponential run-time with respect to the number of jets. This quickly becomes a limiting factor in large datasets, and makes more complex topologies intractable. $\chi^2$ also relies on extensive domain knowledge to construct the functional forms and to eliminate permutations. For example, to minimize the permutation count, it is usual for jets tagged as $b$-jets to be separately permuted, only allowing $b$-tagged jets in $b$-quark positions and vice-versa. However, given that $b$-tagging is not 100% accurate and mis-tags are common, some events become impossible in this formulation.

To our knowledge, the only study of the $t\bar{t}H$ process in which all partons lead to jets which attempts a full event reconstruction is [35], which uses a matrix element method (MEM) to simultaneously reconstruct the event and separate signal and background. Unfortunately, this result does not report any results for the reconstruction efficiency, and the main purpose of the MEM appears to be the signal and background separation rather than the event reconstruction. We are further not aware of any analysis of all-jet $t\bar{t}t\bar{t}$ at all. We thus extend the $\chi^2$ method to $t\bar{t}H$ and $t\bar{t}t\bar{t}$ by adding additional terms to Equation 1. For the Higgs boson in the final state of $t\bar{t}H$, we add a new term incorporating the Higgs mass $m_H = 125$ GeV analogously to how the $W$-boson is included the $t\bar{t}$ case, with $\sigma_H = 22.3$ GeV. For $t\bar{t}t\bar{t}$, we simply modify Equation 1 to have terms for 4 top quarks and 4 $W$-bosons. A complete description of the extended $\chi^2$ methods is available in Appendix D.

We note explicitly that we do not expect the extended $\chi^2$ model to perform well in terms of reconstruction efficiency nor in terms of computation time due to the larger parton and jet multiplicities. We include the extended $\chi^2$ to illustrate the limitations of current methods on

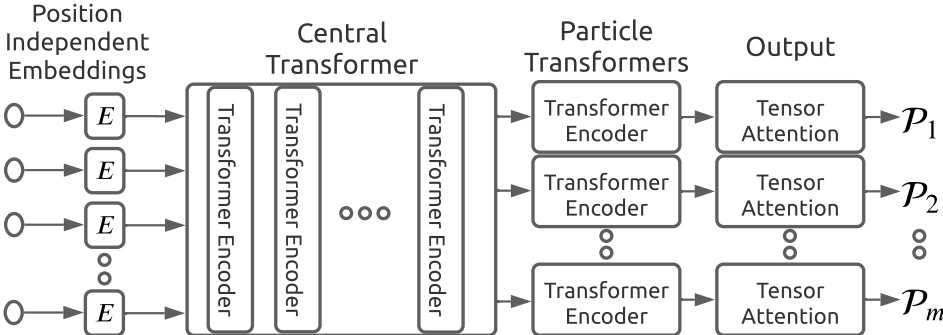

Figure 2: A visualization of the high level structure of SPA-NET.

these larger events. Other methods have been tested for leptonic topologies of $t\bar{t}$, $t\bar{t}H$, or $t\bar{t}t\bar{t}$, such as KLFitter [36], boosted decision trees [31,37,38], and fully connected networks [39]. While these may perform better than the extended $\chi^2$ for $t\bar{t}H$ or $t\bar{t}t\bar{t}$, none have ever been demonstrated to outperform the $\chi^2$ in all-jet $t\bar{t}$. As all of these methods rely on a permutation approach, they are at least as cumbersome and indeed often impossible to work with in a realistic setting, where many millions of events must be evaluated, often hundreds of times due to systematic uncertainties. It is thus beyond the scope of this paper to study the applications of extended permutation techniques for the all-jet channel.

## 3 Symmetry Preserving Attention Networks

We introduce a general architecture for jet-parton assignment named SPA-NET: an attention-based neural network, first described for a specific topology in [40]. In this paper, we generalize the SPA-NET approach from one specific to $t\bar{t}$ to a much more general approach that can accommodate arbitrary event topologies.

**Overview**    The high level structure of SPA-NET, visualized in Figure 2, consists of four distinct components: (1) independent jet embeddings to produce latent space representations for each jet; (2) a central stack of transformer encoders; (3) additional transformer encoders for each particle; and finally (4) a novel tensor-attention to produce the jet-parton assignment distributions. The transformer encoders employ the fairly ubiquitous multi-head self-attention [16]. We replicate the transformer encoder with one modification where we exchange the positional text embeddings with *position-independent* jet embeddings to preserve permutation invariance in the input.

SPA-NET improves run-time performance over baseline permutation methods by avoiding having to construct all valid assignment permutations. Instead, we first partition the jet-parton assignment problem into sub-problems for each resonance particle, as determined by the event Feynman diagram's tree-structure (ex. Figure 1). Then we proceed in two main steps: (1) we solve the jet-parton assignment sub-problems within each of these partitions using a novel form of attention which we call *Symmetric Tensor Attention*; and (2) we combine all the sub-problem solutions into a final jet-parton assignment (Combined Symmetric Loss). This two-step approach also allows us to naturally handle both symmetries described in Section 2.1 within the network architecture.

**Symmetric Tensor Attention**    Every resonance particle $p$ has associated with it $k_p$ partons. Symmetric Tensor Attention takes a set of transformer-encoded jets $X_p \in \mathbb{R}^{N \times D}$ - with $N$ the

total number of jets and $D$ the size of the hidden representation - to produce a rank-$k_p$ tensor $\mathcal{P}_p \in \mathbb{R}^{N \times N \times \cdots \times N}$ such that $\sum \mathcal{P}_p = 1$. $\mathcal{P}_p$ represents a joint distribution over $k_p$-jet assignments indicating the probability that any particular combination of jets is the correct sub-assignment for particle $p$. Additionally, to represent a valid unique solution, all diagonal terms in $\mathcal{P}_p$ must be 0.

We represent jet symmetries (Section 2.1) applicable to the current partition with a *partition-level* permutation group $G_p \subseteq S_{k_p}$ which acts on $k_p$-tuples and defines an equivalence relation over indistinguishable jet assignments. In practice, this equivalence relation is satisfied when the indices of $\mathcal{P}_p$ commute with respect to $G_p$.

$$\forall \sigma \in G_p \ \left( j_1, j_2, \ldots, j_{k_p} \right) \simeq \left( j_{\sigma(1)}, j_{\sigma(2)}, \ldots, j_{\sigma(k_p)} \right) \iff \mathcal{P}_p^{j_1 j_2 \ldots j_{k_p}} = \mathcal{P}_p^{j_{\sigma(1)} j_{\sigma(2)} \ldots j_{\sigma(k_p)}}. \quad (2)$$

We enforce this index commutativity by employing a form of general dot-product attention [12] where the mixing weights mimic the output's symmetries. A Symmetric Tensor Attention (STA) layer contains a single rank-$p_k$ parameter tensor $\Theta \in \mathbb{R}^{D \times D \times \cdots \times D}$ and performs the following computations, expressed using Einstein summation[4] notation:

$$\mathcal{S}^{i_1 i_2 \ldots i_{k_p}} = \sum_{\sigma \in G_p} \Theta^{i_{\sigma(1)} i_{\sigma(2)} \ldots i_{\sigma(k_p)}}, \quad (3)$$

$$\mathcal{O}^{j_1 j_2 \ldots j_{k_p}} = X_{i_1}^{j_1} X_{i_2}^{j_2} \ldots X_{i_{p_k}}^{j_{p_k}} \mathcal{S}^{i_1 i_2 \ldots i_{k_p}}, \quad (4)$$

$$\mathcal{P}_p^{j_1 j_2 \ldots j_{k_p}} = \frac{\exp\left(\mathcal{O}^{j_1 j_2 \ldots j_{k_p}}\right)}{\sum_{j_1, j_2, \ldots, j_{p_k}} \exp\left(\mathcal{O}^{j_1 j_2 \ldots j_{k_p}}\right)}. \quad (5)$$

STA first constructs a $G_p$-symmetric tensor $\mathcal{S}$: the sum over all $G_p$-equivalent indices of $\Theta$ (Equation 3). This is sufficient to ensure the output's indices will also be $G_p$-symmetric (Proof in Appendix A). Afterwards, STA performs a generalized dot-product attention which represents all $k_p$-wise similarities in the input sequence (Equation 4). This operation effectively extends pair-wise general dot-product attention [12] to higher order relationships. This is the most expensive operation in network, with a time and space complexity of $O(N^{k_p})$ (Proof in Appendix A). At this stage, we also mask all diagonal terms in $\mathcal{O}$ by setting them to $-\infty$, enforcing assignment uniqueness. Finally, STA normalizes the output tensor $\mathcal{O}$ by performing a $k_p$-dimensional softmax, producing a final joint distribution $\mathcal{P}_p$ (Equation 5).

**Combined Symmetric Loss** The symmetric attention layers produce solutions $\{\mathcal{P}_1, \mathcal{P}_2, \ldots, \mathcal{P}_m\}$ for each particle's jet-parton assignment sub-problem. The true sub assignments targets for each particle are provided as $\delta$-distributions containing one possible valid jet assignment $\{\mathcal{T}_1, \mathcal{T}_2, \ldots, \mathcal{T}_m\}$. The loss for each sub-problem is simply the cross entropy, $CE(\mathcal{P}_p, \mathcal{T}_p)$, for each particle $p$. We represent particle symmetries (Section 2.1) using an *event-level* permutation group $G_E \subseteq S_m$ and a symmetrized loss. $G_E$ induces an equivalence relation over particles in a manner similar to Equation 2: $\forall \sigma \in G_E, (\mathcal{T}_1, \mathcal{T}_2, \ldots, \mathcal{T}_m) \simeq (\mathcal{T}_{\sigma(1)}, \mathcal{T}_{\sigma(2)}, \ldots, \mathcal{T}_{\sigma(m)})$. This effectively allows us to freely swap entire particles as long as each sub-problem remains correctly assigned.

We incorporate these symmetries into the loss function by allowing the network to fit to *any* equivalent jet assignment, which is achieved by fitting to the minimum attainable loss within a given equivalence class. We also experiment with an alternative loss using soft min

---

[4]It is further worth noting that most linear algebra libraries include an Einstein summation operation (`einsum` [41]) which can efficiently perform this computation on arbitrary tensors.

(Appendix B) to avoid discontinuous behavior.

$$\mathcal{L}_{min} = \min_{\sigma \in G_E} \sum_{i=1}^{m} CE(\mathcal{P}_i, \mathcal{T}_{\sigma(i)}).$$

**Reconstruction** During inference, we generate a final jet-parton assignment by selecting the most likely assignment from each partition's predicted distribution $\mathcal{P}_p$. In the event that we assign a jet to more than one parton, we select the higher probability assignment first and re-evaluate the remaining $\mathcal{P}$'s to select the best non-contradictory assignments. This ensures that our final assignment conforms to the set-assignment uniqueness constraints. This ad-hoc assignment process presents a potential limitation and alternative, more robust approaches may be explored in the future.

## 3.1 Partial Event Reconstruction

Though each parton is usually expected to produce a jet, one or more of these may sometimes not be detected, causing some particles to be impossible to reconstruct. This may be due to limited detector acceptance, merging jets, or other idiosyncrasies. The more partons in the event, the higher the probability that one or more of the particles will be missing a jet. Limiting our dataset to only complete events significantly reduces the available training examples in complex event configurations.

Baseline permutation methods struggle with partial events because their scoring functions are typically only valid for full permutations. Due to SPA-NET's partitioned approach to jet-parton assignment, we can modify our loss to recover any particles which *are* present in the event and still provide a meaningful training signal from these *partial events*. This not only reduces the required training dataset size, but also may reduce generalization bias because such events occur in real collision data.

We mark particles in an event with a masking value $\mathcal{M}_p \in \{0,1\}$ and we only include the loss contributed by reconstructable particles, commuting the target distributions $\mathcal{T}_p$ and masks $\mathcal{M}_p$ together according to $G_E$. We find that the training dataset does not have an equal proportion of all particles, so this masked loss could bias the network towards more common configurations. To accommodate this, we scale the loss based on the distribution of events present in the training dataset by computing the effective class count for each partial combination $CB(\mathcal{M}_1, \mathcal{M}_2, \ldots, \mathcal{M}_m)$ [42] (Appendix B).

$$\mathcal{L}_{min}^{masked} = \min_{\sigma \in G_E} \left( \sum_{i=1}^{m} \frac{\mathcal{M}_{\sigma(i)} CE(\mathcal{P}_i, \mathcal{T}_{\sigma(i)})}{CB\left(\mathcal{M}_{\sigma(1)}, \mathcal{M}_{\sigma(2)}, \ldots, \mathcal{M}_{\sigma(m)}\right)} \right). \tag{6}$$

## 4 Experiments

**Datasets** All processes are generated at a centre-of-mass energy of 13 TeV using MadGraph_aMC@NLO [43] (v2.7.2, NCSA license) for the hard process, Pythia8 [44] (v8.2, GPL-2) for parton showering / hadronisation, and Delphes [45] (v3.4.1, GPL-3) with the AT-LAS parameterization for detector simulation. All $W$-bosons are forced to decay to a pair of light quarks, and Higgs Bosons are forced to decay to $b$-quarks. Jets are reconstructed with the anti-$k_\mathrm{T}$ algorithm [46], radius parameter $R = 0.4$, and must have transverse momentum $p_\mathrm{T} \geq 25$ GeV and absolute pseudo-rapidity $|\eta| < 2.5$. A $b$-tagging algorithm, which identifies jets originating from $b$-quarks, is also applied to each jet with $p_\mathrm{T}$-dependent efficiency and mis-tag rates. The 4-vector $(p_\mathrm{T}, \eta, \phi, M)$ of each jet, as well as the boolean result of the $b$-tagging algorithm, are stored to be used as inputs to the networks. Additional input features

may trivially be added. Of particular interest, jet substructure [47] observables may lead to performance improvements, although we leave such studies for future work.

Truth assignments are generated by matching the partons from the MC event record to the reconstructed jets via the requirement $\sqrt{\Delta\eta^2 + \Delta\phi^2} < 0.4$. We choose to label jets exclusively, such that only one parton may be assigned to each jet, in order to ensure a clean truth definition of the *correct* permutation and a fair comparison to the $\chi^2$ baseline[5]. This truth definition is required in order to define the target distributions during training, but not for network inference except to define performance metrics.

For each topology, we require that every event must contain at least as many jets as we expect in the final state, at least two of which must be $b$-tagged. After filtering, we keep 10M, 14.3M, and 5.8M events out of a total 60M, 100M, and 100M events generated for $t\bar{t}$, $t\bar{t}H$, and $t\bar{t}t\bar{t}$ respectively. For each generated dataset, we used 90% of the events for training, 5% for validation and hyperparameter optimization, and the final 5% for testing. To ensure that our models are not biased to simulator-specific information, we also generate an alternative sample of 100K $t\bar{t}$ validation events using MadGraph_aMC@NLO interfaced to Herwig7 [48] (v7.2.2, GPL) for showering and evaluated them with a model trained only on Pythia8 events.

**SPA-NET Training**     For each event topology, SPA-NET's hyperparameters are chosen using the Sherpa hyperparameter optimization library [49]. We train 200 iterations of each network using 2M events sampled from the complete training dataset. We use a Gaussian process to guide parameter exploration. Each parameter-set was trained for 10 epochs for a total optimization time of 3 days per topology. Final hyperparameters for each benchmark problem are provided in Appendix C.

After hyperparameter optimization, each network was trained using four Nvidia GeForce 3090 GPUs for 50 epochs using the *AdamW* optimizer [50] with $L_2$ regularization on all parameter weights. Additionally, to improve transformer convergence, we anneal the learning rate following a cosine schedule [51], performing a warm restart every 10 epochs. Training took a total of 4 to 6 hours depending on topology.

# 5 Performance

**Reconstruction Efficiency**     We measure model performance via *reconstruction efficiency*, the proportion of correctly assigned jets (also called recall or sensitivity). Efficiencies are evaluated on a per-event and a per-particle basis. We use the three benchmark processes defined in Section 2 - $t\bar{t}$, $t\bar{t}H$, and $t\bar{t}t\bar{t}$ - to evaluate SPA-NET on progressively more complex final states. We compare SPA-NET's performance to the $\chi^2$ baseline described in Section 2.3 both inclusively and as a function of the number of jets in each event ($N_{\text{jets}}$). In what follows, *Complete Events* are those events in which all resonance particles are fully truth-matched to detected partons which are fully reconstructable, while *Partial Events* are those events in which at least one but not all resonance particles are reconstructable. The *Event Fraction* is the percentage of total events included in the denominator for the efficiency calculations. *Event Efficiency* is defined as the proportion of events in which all jets associated with reconstructable particles are correctly assigned. We also report the per-particle *Top Quark Efficiency* and *Higgs Efficiency*. All efficiency values are presented for the testing data split.

$t\bar{t}$ **Results**     Benchmark $t\bar{t}$ reconstruction efficiency is presented in Table 1. We found that SPA-NET outperforms the $\chi^2$ method in every category, with efficiencies consistently around 20%

---

[5]Exclusive matching is required for the baseline $\chi^2$ technique, though not for SPA-NET; we leave studies of non-exclusive matching (so called "boosted" events) to future work.

higher with overall performance on all events increasing from 39.2% to 58.6%. As expected, efficiencies drop off as $N_{\text{jets}}$ increases, and are generally higher in Complete Events than All Events.

Table 1: Performance on for $t\bar{t}$ reconstruction. *Complete Events* contain all resonance particles fully truth-matched to detected partons. *All Events* include complete events as well as partial events.

|  | $N_{\text{jets}}$ | Event Fraction | SPA-NET Efficiency | | $\chi^2$ Efficiency | |
|---|---|---|---|---|---|---|
|  |  |  | Event | Top Quark | Event | Top Quark |
| All Events | == 6 | 0.245 | 0.643 | 0.696 | 0.424 | 0.484 |
|  | == 7 | 0.282 | 0.601 | 0.667 | 0.389 | 0.460 |
|  | $\geq 8$ | 0.320 | 0.528 | 0.613 | 0.309 | 0.384 |
|  | **Inclusive** | **0.848** | **0.586** | **0.653** | **0.392** | **0.457** |
| Complete Events | == 6 | 0.074 | 0.803 | 0.837 | 0.593 | 0.643 |
|  | == 7 | 0.105 | 0.667 | 0.754 | 0.413 | 0.530 |
|  | $\geq 8$ | 0.145 | 0.521 | 0.662 | 0.253 | 0.410 |
|  | **Inclusive** | **0.325** | **0.633** | **0.732** | **0.456** | **0.552** |

$t\bar{t}H$ **Results** $t\bar{t}H$ reconstruction efficiency is presented in Table 2. Note that while SPA-NET is trained on events with $\geq 2$ $b$-tagged jets, the $\chi^2$ method is intractable in this region, due to the additional ambiguities which generate more permutations. Therefore, we compare the two methods only in the subset of events with $\geq 4$ $b$-jets. The $\chi^2$ reconstruction efficiency is extremely low, reaching a maximum event efficiency of just 1.6% on complete events where only 8 truth-matched jets are present. For comparison, SPA-NET achieves 53.2% efficiency in these events. SPA-NET performance in $\geq 2$ $b$-jet events is similar to the $\geq 4$ region (Appendix E); this demonstrates an another advantage of SPA-NET, which can be trained on a more inclusive event selection, reducing the required amount of generated data while still maintaining performance.

Table 2: Performance comparison for $t\bar{t}H$ reconstruction in events with $\geq 4$ $b$-tagged jets.

|  | $N_{\text{jets}}$ | Event Fraction | SPA-NET Efficiency | | | $\chi^2$ Efficiency | | |
|---|---|---|---|---|---|---|---|---|
|  |  |  | Event | Higgs | Top | Event | Higgs | Top |
| All Events | == 8 | 0.261 | 0.370 | 0.497 | 0.540 | 0.044 | 0.151 | 0.053 |
|  | == 9 | 0.313 | 0.343 | 0.492 | 0.514 | 0.038 | 0.146 | 0.066 |
|  | $\geq 10$ | 0.313 | 0.294 | 0.472 | 0.473 | 0.030 | 0.135 | 0.072 |
|  | **Inclusive** | **0.972** | **0.330** | **0.485** | **0.502** | **0.039** | **0.146** | **0.062** |
| Complete Events | == 8 | 0.042 | 0.532 | 0.657 | 0.663 | 0.016 | 0.151 | 0.063 |
|  | == 9 | 0.070 | 0.422 | 0.601 | 0.596 | 0.013 | 0.146 | 0.076 |
|  | $\geq 10$ | 0.115 | 0.306 | 0.545 | 0.523 | 0.008 | 0.134 | 0.080 |
|  | **Inclusive** | **0.228** | **0.383** | **0.583** | **0.572** | **0.012** | **0.144** | **0.073** |

$t\bar{t}t\bar{t}$ **Results** SPA-NET $t\bar{t}t\bar{t}$ reconstruction efficiency is presented Table 3. We do not show results for the $\chi^2$ in this case because the CPU time required simply made it intractable to calculate sufficient statistics for this problem - an event with $N_{\text{jets}} = 12$ and 4 $b$ jets, the simplest possible case, must calculate 2520 permutations, increasing to 113400 for $N_{\text{jets}} = 14$ and 5 $b$ jets. In the extremely limited statistics that were run, performance was close to or

precisely zero. SPA-NET correctly reconstructs 35.0% of the $N_{\text{jets}}$=12 complete events, with a top efficiency of 61.7%. Inclusively, an impressive 19.1% event reconstruction and 52.9% top reconstruction efficiency is achieved despite the huge complexity of the problem. The performance on this dataset emphasizes the importance of the partial-event training approach introduced in Section 3.1, given that only 6.6% of all the training samples were full events. This level of performance even in one of the most complex Standard Model processes currently being analysed at the LHC is an encouraging sign that SPA-NET can handle anything that is given to it without being computationally limited for the forseeable future.

Table 3: Performance of SPA-NET for $t\bar{t}t\bar{t}$ reconstruction in events with $\geq 4$ $b$-tagged jets. No comparison is shown with the $\chi^2$ method, which is intractable in this dataset.

|  | $N_{\text{jets}}$ | Event Fraction | SPA-NET Efficiency Event | SPA-NET Efficiency Top Quark |
|---|---|---|---|---|
| All Events | == 12 | 0.219 | 0.276 | 0.484 |
|  | == 13 | 0.304 | 0.247 | 0.474 |
|  | $\geq$ 14 | 0.450 | 0.198 | 0.450 |
|  | **Inclusive** | **0.974** | **0.231** | **0.464** |
| Complete Events | == 12 | 0.005 | 0.350 | 0.617 |
|  | == 13 | 0.016 | 0.249 | 0.567 |
|  | $\geq$ 14 | 0.044 | 0.149 | 0.504 |
|  | **Inclusive** | **0.066** | **0.191** | **0.529** |

**Computational Overhead**  Figure 3 shows the average evaluation time per event for each benchmark topology, as a function of $N_{\text{jets}}$, for the $\chi^2$ method as well as SPA-NET evaluated on both a CPU and GPU. SPA-NET represents an exponential improvement in run-time on larger events, reducing the $\mathcal{O}(N_{\text{jets}}^C)$, where C is the number of partons, run-time of the $\chi^2$ method to a $\mathcal{O}(N_{\text{jets}}^3)$ across all of our benchmark problems. We also notice an additional factor of 10 improvement when using a GPU for inference as opposed to a CPU. At the LHC, typical dataset sizes are regularly into the tens of millions of events, and it is common to evaluate each of these datasets hundreds of times to evaluate systematic uncertainties. The planned high-luminosity upgrade of the LHC [52] will lead to datasets several orders of magnitude larger, with events

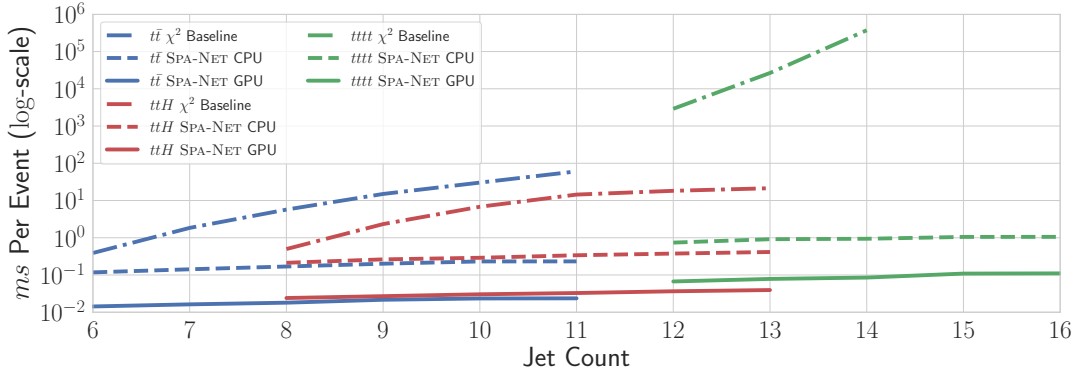

Figure 3: Average run-time for jet-assignment inference of SPA-NET and $\chi^2$ on various events and jet multiplicities over 1000 runs. SPA-NETs were evaluated with a batch size of 1024 events. Timings are performed on an Intel I7 10700K CPU with 64 GB of RAM and Nvidia 3080 GPU.

containing more jets on average [53]. It is thus clear the permutation techniques currently in use do not scale into the future.

**Partial event training** To quantify the improvement in data efficiency when including partial events (Section 3.1), we compare validation efficiency with respect to dataset size. We train on increasingly larger samples from the total generated 14M $t\bar{t}H$ events with and without partial events included (Figure 4b). We notice a statistically significant increase of $\sim 2.5\%$ on all event efficiency in large datasets, increasing to over 5% in small datasets. Additionally, we notice no degradation in complete event efficiency when including partial events. With some LHC analyses already limited by the experimental collaborations' ability to generate sufficient simulated data [37, 54], and casting an eye to future, higher luminosity runs of the LHC, improvements like this are critical.

**Ablation Study** We also evaluate the effect of several smaller aspects of the network on overall reconstruction efficiency. We compare the effect of cosine annealing (Section 4) and several loss modifications such as soft min (Appendix B), effective-count scaling (Appendix B), and partial event training (Section 3.1). In order to estimate statistical uncertainty, each experiment uses random 50% samples of the complete dataset, repeating every experiment with 8 separate samples for each modification. Figure 4a displays the effect of each of these modifications. The effect is small but significant for each considered aspect, with the exception of the partial event training which hugely improves performance when considering all events.

**Simulator Dependence** We check for training bias due to the choice of the MC generators used by evaluating a $t\bar{t}$ SPA-NET trained on a Pythia8 sample on an independent sample generated by Herwig7 (Section 4). Comparisons like these are often used in LHC analyses to assess systematic uncertainties on signal modelling, and indeed it is common to find these systematics among the largest considered even when using non-ML models [37, 55]. We find no degradation in reconstruction efficiency when evaluated on the alternative sample. In fact, both the $\chi^2$ and SPA-NET perform marginally better on the Herwig7 sample, by around 2-6%, as shown in Table 4. We found that on average Herwig7 generates events with fewer jets of

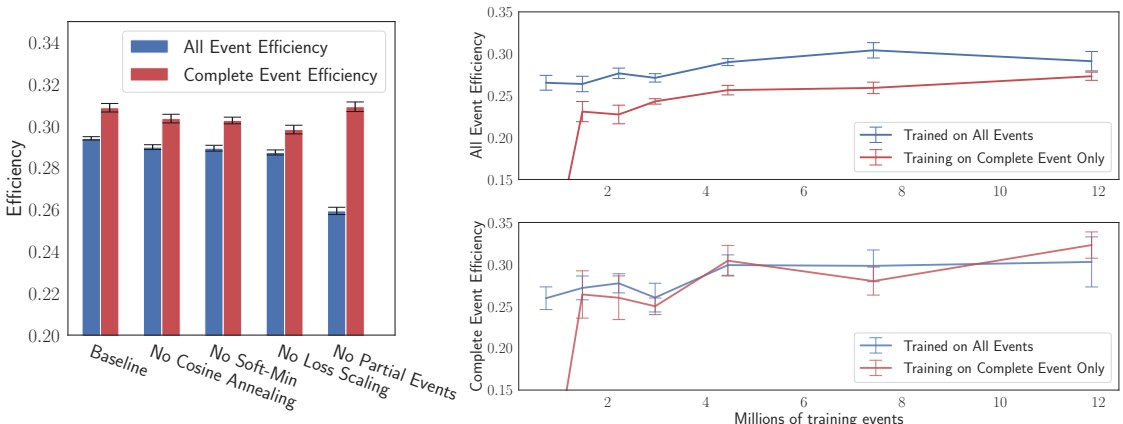

(a) A comparison of different SPA-NET modifications on 6M $t\bar{t}H$ events

(b) Comparison of SPA-NET evaluation efficiencies for various training dataset sizes.

Figure 4: Ablation study efficiencies for both all and complete $t\bar{t}H$ events. *Complete Events* contain all resonance particles fully truth-matched to detected partons. *All Events* include complete events as well as partial events. Error bars indicate 95% confidence intervals over 8 training set samples.

lower $p_T$, which may explain this. Since both approaches improve identically, we do not find training bias introduced from our choice of parton showering package. This is an encouraging indication that performance on real data from the ATLAS or CMS detectors will be similarly unbiased by training on simulated samples, though of course this requires further study by the collaborations.

Table 4: Comparison of results on $t\bar{t}$ using Pythia and Herwig showering.

| | $N_{\text{jets}}$ | Pythia8 | | | | Herwig7 | | | |
| --- | --- | --- | --- | --- | --- | --- | --- | --- | --- |
| | | $\chi^2$ | | SPA-NET | | $\chi^2$ | | SPA-NET | |
| | | Event | Top | Event | Top | Event | Top | Event | Top |
| All Events | $== 6$ | 0.423 | 0.483 | 0.643 | 0.696 | 0.475 | 0.526 | 0.659 | 0.711 |
| | $== 7$ | 0.384 | 0.455 | 0.601 | 0.667 | 0.411 | 0.464 | 0.629 | 0.678 |
| | $\geq 8$ | 0.293 | 0.381 | 0.528 | 0.613 | 0.308 | 0.359 | 0.564 | 0.618 |
| | **Inclusive** | **0.385** | **0.453** | **0.586** | **0.653** | **0.434** | **0.485** | **0.620** | **0.672** |
| Complete Events | $== 6$ | 0.592 | 0.642 | 0.803 | 0.837 | 0.633 | 0.680 | 0.819 | 0.854 |
| | $== 7$ | 0.408 | 0.522 | 0.667 | 0.754 | 0.393 | 0.522 | 0.672 | 0.765 |
| | $\geq 8$ | 0.253 | 0.410 | 0.521 | 0.662 | 0.219 | 0.384 | 0.533 | 0.684 |
| | **Inclusive** | **0.442** | **0.542** | **0.633** | **0.732** | **0.503** | **0.591** | **0.690** | **0.777** |

# 6 Codebase

```
[SOURCE]
mass = log_normalize
pt = log_normalize
eta = normalize
phi = normalize
btag = none

[EVENT]
particles = (t1, t2, H)
permutations = [(t1, t2)]

[t1]
jets = (q1, q2, b)
permutations = [(q1, q2)]

[t2]
jets = (q1, q2, b)
permutations = [(q1, q2)]

[H]
jets = (b1, b2)
permutations = [(b1, b2)]
```

Figure 5: An example config file from the SPA-NET codebase for the $t\bar{t}H$ topology.

In order to minimise the overhead required for independent groups to verify our results and

train their own SPA-NET's, we have released a public codebase under a BSD-3 license[6]. This code can generate network architectures for arbitrary event topologies from a simple config file. As an example, the config used for the $t\bar{t}H$ network is shown in Figure 5.

The first section, titled `[SOURCE]`, lists the per-jet input variables as well as the pre-processing which should be applied. The pre-processing options are `normalize`, `log-normalize`, and `none`, which each operate as the name implies. After that is the `[EVENT]` section, which first defines the resonance particles in the topology. The labels given to these particles are arbitrary, as long as they are used consistently through the config. This section also defines which of these have event-level symmetries, such as the interchangeability of the top and anti-top predictions. Finally, there is a section for each of the resonance particles defined in the previous section, defining the decays of these particles and any jet symmetries between the decay products.

From here, scripts are provided to run training and evaluation. The dependencies of the package are minimal, and a pre-compiled docker image with all necessary libraries installed is provided. The only requirement is that of the input dataset format, an example of which is provided to get users started in preparing their own data.

# 7 Conclusions

We have introduced SPA-NET, a network architecture based on a novel attention mechanism with embedded symmetries, that performs set assignment tasks in a highly effective and efficient manner. We have further released a BSD-3 licensed python package for SPA-NET which can generate appropriate architectures for arbitrary topologies given a user-provided configuration file. We have presented three benchmark use cases of varying complexity from the world of particle physics that demonstrate significantly improved performance, both in terms of the proficiency to predict the correct assignments as well as the computational overhead. Crucially, the computational overhead scales more efficiently with the complexity of the problem when compared to existing benchmark algorithms which quickly becomes intractable. Applications of SPA-NET are not limited to the specific benchmarks we have presented, and the techniques may be generalized to many other LHC use-cases. We have further developed novel techniques which reduce the amount of required training data relative to how neural network training is usually performed in high energy particle physics, something that is crucial as the volume of data from the LHC and associated simulation requirements will continue to grow exponentially in the coming years. All of these developments combined make new analyses tractable for the first time, and may thus be crucial in the discovery of new physics in the LHC era and beyond.

## Acknowledgements

DW and MF are supported by DOE grant DE-SC0009920. S.-C. Hsu is supported by DOW Award DE-SC0015971. AS and PB are in part supported by grants NSF NRT 1633631 and ARO 76649-CS to PB. T.-W.H. is supported by Taiwan MoST grant MOST-107-2112- M-007-029-MY3.

---

[6]Our code can be found at https://github.com/Alexanders101/SPANet

# A   Symmetric Tensor Attention Proofs

**Theorem A.1.** *Given a permutation group $G \subseteq S_k$ for any integer $k$, a rank-$k$ parameter tensor $\Theta \in \mathbb{R}^{D \times D \times \cdots \times D}$, and a set of input vectors $X \in \mathbb{R}^{N \times D}$ the following set of operations*

$$\mathcal{S}^{i_1 i_2 \ldots i_k} = \sum_{\sigma \in G} \Theta^{i_{\sigma(1)} i_{\sigma(2)} \ldots i_{\sigma(k)}}, \tag{7}$$

$$\mathcal{O}^{j_1 j_2 \ldots j_k} = X^{j_1}_{i_1} X^{j_2}_{i_2} \ldots X^{j_k}_{i_k} \mathcal{S}^{i_1 i_2 \ldots i_k}, \tag{8}$$

$$\mathcal{P}^{j_1 j_2 \ldots j_k} = \frac{\exp\left(\mathcal{O}^{j_1 j_2 \ldots j_k}\right)}{\sum_{j_1, j_2, \ldots, j_k} \exp\left(\mathcal{O}^{j_1 j_2 \ldots j_k}\right)}, \tag{9}$$

*will produce a an output tensor, $\mathcal{P}$, that is $G$-symmetric. That is,*

$$\forall \sigma \in G, \ \mathcal{P}^{j_1 j_2 \ldots j_k} = \mathcal{P}^{j_{\sigma(1)} j_{\sigma(2)} \ldots j_{\sigma(k)}}.$$

*Proof.* In order to prove that the output, $\mathcal{P}$, is $G$-symmetric, it is sufficient to prove that every step produces a $G$-symmetric tensor. We will now prove that the result from all three steps will be $G$-symmetric.

- We will first prove that the output to Equation 7, which is known as the (unnormalized) *symmetric part* of tensor $\Theta$, will be $G$-symmetric. That is,

$$\forall \tau \in G, \ \mathcal{S}^{j_1 j_2 \ldots j_k} = \mathcal{S}^{j_{\tau(1)} j_{\tau(2)} \ldots j_{\tau(k)}}.$$

Since $G$ is a group, for every element $\nu \in G$, there exists a unique $\sigma \in G$ such that $\nu = \sigma\tau$. This is a consequence of the unique inverse property of groups, forcing that element to be $\sigma = \nu\tau^{-1}$. Therefore,

$$
\begin{aligned}
\mathcal{S}^{j_{\tau(1)} j_{\tau(2)} \ldots j_{\tau(k)}} &= \sum_{\sigma \in G} \Theta^{i_{\sigma(\tau(1))} i_{\sigma(\tau(2))} \ldots i_{\sigma(\tau(k))}} \\
&= \sum_{\nu \in G} \Theta^{i_{(\nu\tau^{-1}\tau)(1)} i_{(\nu\tau^{-1}\tau)(2)} \ldots i_{(\nu\tau^{-1}\tau)(k)}} \\
&= \sum_{\nu \in G} \Theta^{i_{\nu(1)} i_{\nu(2)} \ldots i_{\nu(k)}} \\
&= \mathcal{S}^{j_1 j_2 \ldots j_k}.
\end{aligned}
$$

- For Equation 8, we use the same $X$ tensor $k$ times in the expression. Since these tensors are all identical, they are trivially symmetric since and we can freely swap the order of the $X$ tensors as long as we apply an inverse permutation to another set of indices. Furthermore, since $\mathcal{S}$ is $G$-symmetric from the previous step, it can also freely permute its indices according to $G$. Therefore,

$$
\begin{aligned}
\forall \sigma \in G, \mathcal{O}^{j_{\sigma(1)} j_{\sigma(2)} \ldots j_{\sigma(k)}} &= X^{j_{\sigma(1)}}_{i_1} X^{j_{\sigma(2)}}_{i_2} \ldots X^{j_{\sigma(k)}}_{i_k} \mathcal{S}^{i_1 i_2 \ldots i_k} \\
&= X^{j_1}_{i_{\sigma^{-1}(1)}} X^{j_2}_{i_{\sigma^{-1}(2)}} \ldots X^{j_k}_{i_{\sigma^{-1}(k)}} \mathcal{S}^{i_1 i_2 \ldots i_k} \\
&= X^{j_1}_{i_1} X^{j_2}_{i_2} \ldots X^{j_k}_{i_k} \mathcal{S}^{i_{\sigma(1)} i_{\sigma(2)} \ldots i_{\sigma(k)}} \\
&= X^{j_1}_{i_1} X^{j_2}_{i_2} \ldots X^{j_k}_{i_k} \mathcal{S}^{i_1 i_2 \ldots i_k} \\
&= \mathcal{O}^{j_1 j_2 \ldots j_k}.
\end{aligned}
$$

- For Equation 9, the operations are performed element-wise to every element in $\mathcal{O}$ and the normalisation term is simply the sum of all elements in $\exp(\mathcal{O})$. Since summation is commutative,

$$\forall \sigma \in G, \sum_{j_1, j_2, \ldots, j_k} \exp\left(\mathcal{O}^{j_1 j_2 \ldots j_k}\right) = \sum_{j_{\sigma(1)}, j_{\sigma(2)}, \ldots, j_{\sigma(k)}} \exp\left(\mathcal{O}^{j_{\sigma(1)} j_{\sigma(2)} \ldots j_{\sigma(k)}}\right).$$

Combining the fact that both the normalisation term and $\mathcal{O}$ are both $G$-symmetric, we find that the output is also $G$-symmetric.

$$\forall \sigma \in G, \ \mathcal{P}^{j_{\sigma(1)} j_{\sigma(2)} \ldots j_{\sigma(k)}} = \frac{\exp\left(\mathcal{O}^{j_{\sigma(1)} j_{\sigma(2)} \ldots j_{\sigma(k)}}\right)}{\sum_{i_{\sigma(1)}, i_{\sigma(2)}, \ldots, i_{\sigma(k)}} \exp\left(\mathcal{O}^{i_{\sigma(1)} i_{\sigma(2)} \ldots i_{\sigma(k)}}\right)}$$
$$= \frac{\exp\left(\mathcal{O}^{j_1 j_2 \ldots j_k}\right)}{\sum_{i_1, i_2, \ldots, i_k} \exp\left(\mathcal{O}^{i_1 i_2 \ldots i_k}\right)}$$
$$= \mathcal{P}^{j_1 j_2 \ldots j_k}.$$

$\square$

## A.1 Run-time Complexity

For the following sections, we will treat the network's hidden representation dimension $D$ as a constant. This is because this value is a hyperparameter which may be adjusted freely, although we also provide the run-time expressions with $D$ present.

- Equation 7. This expression is simply an element-wise sum over all possible elements of group $G$ and tensor $\Theta$. The run-time of this step is therefore exponential with respect to the number of partons in each partition.

$$O\left(|G|D^k\right) = O\left(k! D^k\right)$$
$$= O\left(k^k D^k\right)$$
$$= O\left((Dk)^k\right)$$
$$= O\left(k^k\right).$$

- Equation 8. This expression evaluates a generalized tensor-product between $k$ rank-2 tensors and one rank-$k$ tensor. The output will be rank-$k$ tensor with sizes $N \times N \times \cdots \times N$. For each of these outputs, the operation must perform a rank-$k$ tensor multiplication with sizes $D \times D \times \cdots \times D$. The run-time of this step is therefore exponential with respect to the number of partons in each partition. We note that this is only the naive run-time and many tensor-multiplication libraries will not use divide-and-conquer algorithms to reduce the $O\left(D^k\right)$ multiplication operation.

$$O\left(N^k D^k\right) = O\left((ND)^k\right)$$
$$= O\left(N^k\right).$$

- Equation 9. The normalization factor can be pre-computed once for every element of $\mathcal{O}$. This expression then reduces to simply an element-wise exponentiation and division over all $O\left(N^k\right)$ elements in $\mathcal{O}$

The total run-time complexity of the symmetric tensor attention layer assuming that $D$ is constant is therefore simply

$$O(k^k + N^k).$$

# B    SPANet Modifications

## B.1    Soft Loss Function

When constructing the symmetric loss function, we use the minimum loss over all equivalent particle orderings as our optimization objective. However, this might cause instability on events where the network is unsure, causing the loss function to flip every epoch for that event. In order to prevent this and maintain a continually differentiable loss function, we use a modified loss based on the soft min function.

$$\mathcal{L}_{softmin} = \text{soft}\min_{\sigma \in G_E} \sum_{i=1}^{m} CE(\mathcal{P}_i, \mathcal{T}_{\sigma(i)}),$$

where

$$\text{soft}\min\{x_1, x_2, \ldots, x_k\} = \sum_{j=1}^{k} \frac{e^{-x_j}}{\sum_{i=1}^{k} e^{-x_i}} x_j.$$

## B.2    Balanced Loss Scaling

We experiment with balancing the loss based on the prevalence of each combination of particles in the target set. This is primarily to prevent the network from ignoring rare events such as the complete $t\bar{t}t\bar{t}$ event when performing partial event training. If there is a large imbalance between classes, such as when events with fewer particle are more prevalent, this could cause the network to bias its results towards those more common events and worsen performance on full events.

We compute the class balance term $CB(\mathcal{M}_1, \mathcal{M}_2, \ldots, \mathcal{M}_m)$ where the $\mathcal{M}_p$ terms represent binary values indicating if a particle $p$ is present or not in the event and $m$ is the total number of particles. If $\mathcal{M}_p = 1$, then $p$ is fully reconstructable in the given event, and if $\mathcal{M}_p = 0$, then at least one parton associated with particle $p$ is not detectable.

Assume we have a dataset of size $N$ of such events, each with their own masking vector for each possible particle $\mathcal{M}_p^j$ for $1 \leq j \leq N$ and $1 \leq p \leq m$. We will keep the particle indices in the subscript and the dataset indices in the superscript. Assume we also have an *event-level* permutation group $G_E \subseteq S_m$ (Section 3). We define $CB$ based on *effective class count* [42].

First, we will define a counting function. Let $\mathbb{1}_P$ be the selection function for prediction $P$. This is,

$$\mathbb{1}_P = \begin{cases} 1 & \text{if } P \text{ is } True \\ 0 & \text{Otherwise} \end{cases}.$$

Next, define label-counting function $C$ which simply counts how many times a particular arrangement of masking values appears in our dataset.

$$C(\mathcal{M}_1, \mathcal{M}_2, \ldots, \mathcal{M}_m) = \sum_{j=1}^{N} \prod_{p=1}^{m} \mathbb{1}_{\mathcal{M}_p^j = \mathcal{M}_p}.$$

Such a counting function does not account for the equivalent particle assignments that are induced by our event-level group $G_E$. To accommodate particle symmetries, we create a symmetric counting function $S$ which counts not only the presence of any particular arrangement of masking values, but also all *equivalent* arrangements.

$$S(\mathcal{M}_1, \mathcal{M}_2, \ldots, \mathcal{M}_m) = \sum_{\sigma \in G_E} \sum_{j=1}^{N} \prod_{p=1}^{m} \mathbb{1}_{\mathcal{M}_p^j = \mathcal{M}_{\sigma(p)}}.$$

Notice that this definition guarantees that any two equivalent masking value sets will have identical symmetric class counts.

$$\forall \sigma \in G_E, \; S(\mathcal{M}_1, \mathcal{M}_2, \ldots, \mathcal{M}_m) = S(\mathcal{M}_{\sigma(1)}, \mathcal{M}_{\sigma(2)}, \ldots, \mathcal{M}_{\sigma(m)}).$$

We set the scale $\beta$ in effective class definition based on the size of our dataset $N$.

$$\beta = 1 - 10^{-\log_{10} N}.$$

Finally, We define the class balance ($CB$) as the normalized values of the effective class counts ($ECC$) [42]

$$ECC(\mathcal{M}_1, \mathcal{M}_2, \ldots, \mathcal{M}_m) = \frac{1 - \beta^{S(\mathcal{M}_1, \mathcal{M}_2, \ldots, \mathcal{M}_m)}}{1 - \beta},$$

$$CB(\mathcal{M}_1, \mathcal{M}_2, \ldots, \mathcal{M}_m) = \frac{ECC(\mathcal{M}_1, \mathcal{M}_2, \ldots, \mathcal{M}_m)}{\sum_{\mathcal{M}' \in \{0,1\}^m} ECC(\mathcal{M}'_1, \mathcal{M}'_2, \ldots, \mathcal{M}'_m)}.$$

## C Hyperparameters

Table 5: A complete table of all hyper-parameters used during SPA-NET training on all benchmark problems.

| Parameter | Benchmark Problems | | |
|---|---|---|---|
| | $t\bar{t}$ | $t\bar{t}H$ | $t\bar{t}t\bar{t}$ |
| Training Epochs | 50 | 50 | 50 |
| Learning Rate | 0.0015 | 0.00302 | 0.0015 |
| Batch Size | 2048 | 2048 | 2048 |
| Dropout Percentage | 0.1 | 0.1 | 0.1 |
| $L_2$ Gradient Clipping | N/A | 0.1 | N/A |
| $L_2$ Weight Normalization | 0.0002 | 0.0000683 | 0.0002 |
| Hidden Dimensionality | 128 | 128 | 128 |
| Central Encoder Count | 6 | 5 | 2 |
| Branch Encoder Count | 3 | 5 | 7 |
| Partial Event Training | Yes | Yes | Yes |
| Loss Scaling | Yes | Yes | Yes |
| Loss Type | $\mathcal{L}_{\min}$ | $\mathcal{L}_{\text{soft min}}$ | $\mathcal{L}_{\text{soft min}}$ |
| Cosine Annealing Cycles | 5 | 5 | 5 |

## D $\chi^2$ Method Details

In Section 2.3, we introduce the $\chi^2$ method for reconstructing $t\bar{t}$ events. This is a standard benchmark against which we can compare the results from SPA-NET, and has been used in multiple published results, such as [32, 33]. However, no such benchmark exists for the $t\bar{t}H$ and $t\bar{t}t\bar{t}$ topologies. We thus extend the $\chi^2$ method to these topologies in a simple way in order to have a benchmark to compare against.

The $t\bar{t}$ formulation we use is given in Equation 1. In [40], a different formulation of $\chi^2$ was used that more closely matches recent ATLAS results in which $\sigma_t$ is not used explicitly. While this formulation reduces mass sculpting of incomplete and background events, it does not perform well on events partial events with only a single reconstructable top quark. Further,

it is unclear how to optimally extend this formulation to the $t\bar{t}t\bar{t}$ case. Thus, in this work we prefer the formulation that explicitly includes $m_t$.

The $\chi^2$ is evaluated on $t\bar{t}H$ events as:

$$\chi^2_{t\bar{t}H} = \frac{(m_{b_1q_1q_1} - m_t)^2}{\sigma_t^2} + \frac{(m_{b_2q_2q_2} - m_t)^2}{\sigma_t^2}$$
$$+ \frac{(m_{q_1q_1} - m_W)^2}{\sigma_W^2} + \frac{(m_{q_2q_2} - m_W)^2}{\sigma_W^2} + \frac{(m_{b_0b_0} - m_H)^2}{\sigma_H^2} , \tag{10}$$

where we have simply added an additional term to Equation 1 for the Higgs boson, analogously to the terms used for the $W$-bosons. We label the jets hypothesized to be the decay products of the Higgs boson as $b_0$ here and find $\sigma_H = 22.3$ GeV in our dataset.

The $\chi^2$ for $t\bar{t}t\bar{t}$ is given by the expression

$$\chi^2_{t\bar{t}t\bar{t}} = \frac{(m_{b_1q_1q_1} - m_t)^2}{\sigma_t^2} + \frac{(m_{b_2q_2q_2} - m_t)^2}{\sigma_t^2} + \frac{(m_{b_3q_3q_3} - m_t)^2}{\sigma_t^2} + \frac{(m_{b_4q_4q_4} - t)^2}{\sigma_t^2}$$
$$+ \frac{(m_{q_1q_1} - m_W)^2}{\sigma_W^2} + \frac{(m_{q_2q_2} - m_W)^2}{\sigma_W^2} + \frac{(m_{q_3q_3} - m_W)^2}{\sigma_W^2} + \frac{(m_{q_4q_4} - m_W)^2}{\sigma_W^2} , \tag{11}$$

where we have simply added additional, identical terms for the third and fourth top quarks and $W$-bosons. We find that the complexity of the 12 parton final state makes this effectively intractable and thus do not present reconstruction performance with this formulation, presenting it only as a demonstration that the CPU overhead required in this topology means permutation methods do not scale to these events.

# E   Additional Result Tables

Table 6: SPA-NET results on $t\bar{t}$ using Pythia showering.

| | $N_{\text{jets}}$ | Event Fraction | Event Efficiency | Top Quark Efficiency |
|---|---|---|---|---|
| All Events | == 6 | 0.245 | 0.643 | 0.696 |
| | == 7 | 0.282 | 0.601 | 0.667 |
| | ≥ 8 | 0.320 | 0.528 | 0.613 |
| | **Inclusive** | **0.848** | **0.586** | **0.653** |
| 1 Top Events | == 6 | 0.171 | 0.574 | 0.574 |
| | == 7 | 0.176 | 0.562 | 0.562 |
| | ≥ 8 | 0.175 | 0.534 | 0.534 |
| | **Inclusive** | **0.524** | **0.556** | **0.556** |
| 2 Top Events | == 6 | 0.073 | 0.803 | 0.837 |
| | == 7 | 0.105 | 0.667 | 0.754 |
| | ≥ 8 | 0.144 | 0.521 | 0.662 |
| | **Inclusive** | **0.325** | **0.633** | **0.732** |

Table 7: $\chi^2$ method results on $t\bar{t}$ using Pythia showering.

| | $N_{\text{jets}}$ | Event Fraction | Event Efficiency | Top Quark Efficiency |
|---|---|---|---|---|
| All Events | $== 6$ | 0.245 | 0.424 | 0.484 |
| | $== 7$ | 0.282 | 0.389 | 0.460 |
| | $\geq 8$ | 0.320 | 0.309 | 0.384 |
| | **Inclusive** | **0.848** | **0.392** | **0.457** |
| 1 Top Events | $== 6$ | 0.171 | 0.355 | 0.355 |
| | $== 7$ | 0.176 | 0.373 | 0.373 |
| | $\geq 8$ | 0.175 | 0.348 | 0.348 |
| | **Inclusive** | **0.524** | **0.359** | **0.359** |
| 2 Top Events | $== 6$ | 0.074 | 0.593 | 0.643 |
| | $== 7$ | 0.105 | 0.413 | 0.530 |
| | $\geq 8$ | 0.145 | 0.253 | 0.410 |
| | **Inclusive** | **0.325** | **0.456** | **0.552** |

Table 8: SPA-NET results on $t\bar{t}$ using Herwig showering.

| | $N_{\text{jets}}$ | Event Fraction | Event Efficiency | Top Quark Efficiency |
|---|---|---|---|---|
| All Events | $== 6$ | 0.220 | 0.659 | 0.711 |
| | $== 7$ | 0.222 | 0.629 | 0.678 |
| | $\geq 8$ | 0.185 | 0.564 | 0.618 |
| | **Inclusive** | **0.629** | **0.620** | **0.672** |
| 1 Top Events | $== 6$ | 0.156 | 0.593 | 0.593 |
| | $== 7$ | 0.163 | 0.614 | 0.614 |
| | $\geq 8$ | 0.138 | 0.575 | 0.575 |
| | **Inclusive** | **0.459** | **0.595** | **0.595** |
| 2 Top Events | $== 6$ | 0.064 | 0.819 | 0.854 |
| | $== 7$ | 0.059 | 0.672 | 0.765 |
| | $\geq 8$ | 0.046 | 0.533 | 0.684 |
| | **Inclusive** | **0.170** | **0.690** | **0.777** |

Table 9: $\chi^2$ method results on $t\bar{t}$ using Herwig showering.

| | $N_{\text{jets}}$ | Event Fraction | Event Efficiency | Top Quark Efficiency |
|---|---|---|---|---|
| All Events | $== 6$ | 0.220 | 0.505 | 0.560 |
| | $== 7$ | 0.222 | 0.442 | 0.488 |
| | $\geq 8$ | 0.185 | 0.338 | 0.386 |
| | **Inclusive** | **0.629** | **0.434** | **0.484** |
| 1 Top Events | $== 6$ | 0.156 | 0.434 | 0.434 |
| | $== 7$ | 0.163 | 0.442 | 0.442 |
| | $\geq 8$ | 0.138 | 0.363 | 0.363 |
| | **Inclusive** | **0.459** | **0.415** | **0.415** |
| 2 Top Events | $== 6$ | 0.064 | 0.678 | 0.713 |
| | $== 7$ | 0.059 | 0.442 | 0.553 |
| | $\geq 8$ | 0.046 | 0.263 | 0.419 |
| | **Inclusive** | **0.170** | **0.483** | **0.577** |

Table 10: SPA-NET results on $t\bar{t}H$ with at least 2 $b$-tagged jets (all generated events).

|  | $N_{\text{jets}}$ | Event Fraction | Event Efficiency | Higgs Efficiency | Top Quark Efficiency |
|---|---|---|---|---|---|
| All Events | == 8 | 0.281 | 0.329 | 0.430 | 0.498 |
|  | == 9 | 0.316 | 0.304 | 0.430 | 0.476 |
|  | ≥ 10 | 0.355 | 0.264 | 0.420 | 0.441 |
|  | **Inclusive** | **0.954** | **0.297** | **0.426** | **0.468** |
| Higgs Events | == 8 | 0.197 | 0.317 | 0.430 | 0.531 |
|  | == 9 | 0.227 | 0.295 | 0.430 | 0.504 |
|  | ≥ 10 | 0.261 | 0.257 | 0.420 | 0.462 |
|  | **Inclusive** | **0.686** | **0.287** | **0.426** | **0.493** |
| 1 Top Events | == 8 | 0.167 | 0.314 | 0.413 | 0.466 |
|  | == 9 | 0.177 | 0.297 | 0.409 | 0.448 |
|  | ≥ 10 | 0.184 | 0.273 | 0.397 | 0.421 |
|  | **Inclusive** | **0.529** | **0.294** | **0.406** | **0.444** |
| 2 Top Events | == 8 | 0.066 | 0.352 | 0.590 | 0.539 |
|  | == 9 | 0.092 | 0.295 | 0.540 | 0.504 |
|  | ≥ 10 | 0.130 | 0.225 | 0.490 | 0.456 |
|  | **Inclusive** | **0.289** | **0.277** | **0.526** | **0.490** |
| Full Events | == 8 | 0.036 | 0.440 | 0.590 | 0.599 |
|  | == 9 | 0.057 | 0.344 | 0.540 | 0.542 |
|  | ≥ 10 | 0.087 | 0.248 | 0.490 | 0.480 |
|  | **Inclusive** | **0.180** | **0.317** | **0.526** | **0.523** |

Table 11: SPA-NET results on $t\bar{t}H$ with at least 4 $b$-tagged jets (filtered events).

|  | $N_{\text{jets}}$ | Event Fraction | Event Efficiency | Higgs Efficiency | Top Quark Efficiency |
|---|---|---|---|---|---|
| All Events | == 8 | 0.260 | 0.370 | 0.497 | 0.540 |
|  | == 9 | 0.313 | 0.343 | 0.492 | 0.514 |
|  | ≥ 10 | 0.397 | 0.294 | 0.472 | 0.473 |
|  | **Inclusive** | **0.972** | **0.330** | **0.485** | **0.502** |
| Higgs Events | == 8 | 0.209 | 0.380 | 0.497 | 0.580 |
|  | == 9 | 0.252 | 0.355 | 0.492 | 0.550 |
|  | ≥ 10 | 0.320 | 0.302 | 0.472 | 0.501 |
|  | **Inclusive** | **0.782** | **0.340** | **0.485** | **0.535** |
| 1 Top Events | == 8 | 0.153 | 0.335 | 0.479 | 0.494 |
|  | == 9 | 0.171 | 0.324 | 0.474 | 0.475 |
|  | ≥ 10 | 0.199 | 0.296 | 0.448 | 0.446 |
|  | **Inclusive** | **0.524** | **0.316** | **0.466** | **0.469** |
| 2 Top Events | == 8 | 0.061 | 0.435 | 0.657 | 0.597 |
|  | == 9 | 0.096 | 0.360 | 0.601 | 0.550 |
|  | ≥ 10 | 0.153 | 0.269 | 0.545 | 0.491 |
|  | **Inclusive** | **0.310** | **0.330** | **0.583** | **0.530** |
| Full Events | == 8 | 0.042 | 0.532 | 0.657 | 0.663 |
|  | == 9 | 0.070 | 0.422 | 0.601 | 0.596 |
|  | ≥ 10 | 0.116 | 0.306 | 0.545 | 0.523 |
|  | **Inclusive** | **0.228** | **0.383** | **0.583** | **0.572** |

Table 12: SPA-NET results on $t\bar{t}t\bar{t}$ with at least 2 $b$-tagged jets (all generated events).

|  | $N_{\text{jets}}$ | Event Fraction | Event Efficiency | Top Quark Efficiency |
|---|---|---|---|---|
| All Events | == 12 | 0.227 | 0.257 | 0.458 |
|  | == 13 | 0.309 | 0.232 | 0.453 |
|  | ≥ 14 | 0.433 | 0.185 | 0.426 |
|  | **Inclusive** | **0.970** | **0.217** | **0.441** |
| 1 Top Events | == 12 | 0.060 | 0.412 | 0.412 |
|  | == 13 | 0.069 | 0.399 | 0.399 |
|  | ≥ 14 | 0.073 | 0.374 | 0.374 |
|  | **Inclusive** | **0.202** | **0.394** | **0.394** |
| 2 Top Events | == 12 | 0.106 | 0.217 | 0.441 |
|  | == 13 | 0.136 | 0.206 | 0.430 |
|  | ≥ 14 | 0.172 | 0.181 | 0.406 |
|  | **Inclusive** | **0.415** | **0.198** | **0.423** |
| 3 Top Events | == 12 | 0.056 | 0.162 | 0.482 |
|  | == 13 | 0.089 | 0.148 | 0.471 |
|  | ≥ 14 | 0.148 | 0.117 | 0.436 |
|  | **Inclusive** | **0.294** | **0.135** | **0.455** |
| 4 Top Events | == 12 | 0.005 | 0.297 | 0.580 |
|  | == 13 | 0.014 | 0.211 | 0.543 |
|  | ≥ 14 | 0.039 | 0.111 | 0.470 |
|  | **Inclusive** | **0.059** | **0.152** | **0.497** |

Table 13: SPA-NET results on $t\bar{t}t\bar{t}$ with at least 4 $b$-tagged jets (filtered events).

|  | $N_{\text{jets}}$ | Event Fraction | Event Efficiency | Top Quark Efficiency |
|---|---|---|---|---|
| All Events | == 12 | 0.219 | 0.276 | 0.484 |
|  | == 13 | 0.304 | 0.247 | 0.474 |
|  | ≥ 14 | 0.450 | 0.198 | 0.450 |
|  | **Inclusive** | **0.974** | **0.231** | **0.464** |
| 1 Top Events | == 12 | 0.055 | 0.422 | 0.422 |
|  | == 13 | 0.062 | 0.414 | 0.414 |
|  | ≥ 14 | 0.0684 | 0.388 | 0.388 |
|  | **Inclusive** | **0.185** | **0.407** | **0.407** |
| 2 Top Events | == 12 | 0.101 | 0.235 | 0.461 |
|  | == 13 | 0.132 | 0.222 | 0.445 |
|  | ≥ 14 | 0.175 | 0.194 | 0.420 |
|  | **Inclusive** | **0.410** | **0.213** | **0.438** |
| 3 Top Events | == 12 | 0.057 | 0.200 | 0.513 |
|  | == 13 | 0.094 | 0.172 | 0.492 |
|  | ≥ 14 | 0.162 | 0.136 | 0.460 |
|  | **Inclusive** | **0.313** | **0.159** | **0.479** |
| 4 Top Events | == 12 | 0.006 | 0.350 | 0.617 |
|  | == 13 | 0.016 | 0.249 | 0.567 |
|  | ≥ 14 | 0.044 | 0.149 | 0.504 |
|  | **Inclusive** | **0.066** | **0.191** | **0.529** |

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
