# Peer review of "SPANet: Generalized Permutationless Set Assignment for Particle Physics using Symmetry Preserving Attention"

_SciPost Physics, doi:SciPost Phys. 12, 178 (2022)_

## Round 2 · Referee Report · Anonymous (Referee 1) · 2021-12-10

Strengths

1: Proposes an innovative ML-based method for a set-based assignment problem in particle physics (jet-parton assignment) taking into account permutation symmetries of the system.

2: Provides results on three benchmark problems of increasing complexity together with an analysis of runtime properties. The results show a clear improvement, especially for the most complex problem considered (tttt)

3) As e.g. precision measurements of the Higgs sector increasingly tackle complex final states, the proposed method to resolve the parton assignment is timely and very well motivated. The provided implementation should experimental deployment.

Weaknesses

1) In the context of baseline methods, a discussion of the Matrix Element method would be useful. For example, there exist results by the CMS collaboraton using a matrix element approach for the all-hadronic ttH final state (1803.06986,). The sentence "there exists no study of the ttH¯ process in which all partons lead to jets which attempts a full event reconstruction" should be amended accordingly.

2) Table 2: Why is the matching efficiency using the chi2 method lower for complete events than for all events? Can this be understood in more detail?

Report

Parton assignment is an important aspect of event classification in particle physics, and required for many analyses of complex final states. Especially in future data taking periods of the LHC/HL-LHC, measurements of such topologies will become increasingly relevant.

The paper makes important progress on this challenges by proposing a novel and well-principled machine learning approach to efficiently assign permutations.

We comment on some minor issues above, but once these are resolved fully recommend the manuscript for publication in SciPost.

Requested changes

1) Include discussion of MEM (see above)

2) Consider moving section 6 (Codebase) to an Appendix

  • validity: top
  • significance: high
  • originality: high
  • clarity: good
  • formatting: perfect
  • grammar: perfect

Author:  Michael James Fenton  on 2021-12-23  [id 2052]

(in reply to Report 1 on 2021-12-10)

We have edited the baseline discussion section as follows;

"To our knowledge, there exists no study of the \ttH{} process in which all partons lead to jets which attempts a full event reconstruction, and we are further not aware of any analysis of all-jet \tttt{} at all."

Changed to

"To our knowledge, the only study of the \ttH{} process in which all partons lead to jets which attempts a full event reconstruction is \cite{CMS:2018sah}, which uses a matrix element method (MEM) to simultaneously reconstruct the event and separate signal and background. Unfortunately, this result does not report any results for the reconstruction efficiency, and the main purpose of the MEM appears to be the signal and background separation rather than the event reconstruction. We are further not aware of any analysis of all-jet \tttt{} at all."

2) Table 2: Why is the matching efficiency using the chi2 method lower for complete events than for all events? Can this be understood in more detail?

This appears to be a consequence of the cuts and data distributions within each cut. If you look at the event fractions, the “Complete” events are mostly dominated by high jet-multiplicity events, i.e. more difficult ones. This makes intuitive sense since higher multiplicity events are more likely to be successfully truth matched. The chi^2 performance degrades rapidly vs NJets, reducing efficiency numbers on these more challenging cuts.

It is the preference of the authors to keep section 6 in the main body rather than moving to an appendix, but we are open to this change if the editor prefers it. We have kept it in the main body for this revision.

---

## Round 3 · Referee Report · Anonymous · 2022-1-4

Report

I thank the authors for the additional clarifications and changes to the manuscript and support publication in the current form.

---

## Round 3 · Author Response

Resubmission to address minor editorial comments from reviewer

---

## Round 3 · List of Changes

Added short discussion of MEM from CMS ttH analysis

You are currently on this page

Resubmission 2106.03898v3 on 24 December 2021

---

## Editorial Decision

published